# Use of screens, books and adults' interactions on toddler's language and motor skills: A cross-cultural study among 19 Latin American countries from different SES

Lucas G. Gago-Galvagno[1]*, Angel M. Elgier[2], Angel J. Tabullo[3], Edson J. Huaire-Inacio[4], Angela M. Herrera-Alvarez[4], Carmen Zambrano-Villalba[5], Frania R. López[6], Emmanuel Herrera-González[7], Olivia Morán-Núñez[8], Neyra J. Ochoa-Vega[9], Chrissie Ferreira de Carvalho[10], Rosario Spencer[11], Juan José Giraldo-Huertas[2], Perla del Carpio[12], Erika Robles[13], Carla Fernández[14], Silvia Requena[15], Pilar Rodríguez[16], Yoysy Rondón[17], Alexis L. Ruiz[18], Ada Tibisay-Echenique[19], Gris Hidalgo[20], Carlos R. Hernández[20], Mirna Lacayo[21], Esther Angeriz[22], Gabriela Etchebehere[22], Fernando José Mena[23], Delia Magaña de Ávila[23], Susana C. Azzollini[24], Stephanie E. Miller[25]

1 Facultad de Psicología y Relaciones Humanas, Universidad Abierta Interamericana (UAI), Consejo Nacional de Investigaciones Científicas y Técnicas (CONICET), Buenos Aires, Argentina, 2 Facultad de Psicología y Ciencias del Comportamiento, Universidad de La Sabana, Chía, Colombia, 3 Facultad de Humanidades y Ciencias Económicas, Pontificia Universidad Católica Argentina, Instituto de Ciencias Humanas, Sociales y Ambientales (INCIHUSA), CCT-Mendoza, Consejo Nacional de Investigaciones Científicas y Técnicas (CONICET), Mendoza, Argentina, 4 Universidad César Vallejo, Lima, Perú, 5 Universidad Estatal de Milagro, Milagro, Ecuador, 6 Delegación Municipal de León, Ministerio de Educación (MINED), León, Nicaragua, 7 Programa PSICOMI, Escuela de Ciencias del Movimiento Humano y Calidad de Vida, Universidad Nacional, Heredia, Costa Rica, 8 Universidad de Panamá, Facultad de Psicología, Centro Regional Universitario de Veraguas, Ciudad de Santiago, Panamá, 9 Universidad de Panamá, Facultad de Psicología, Centro Regional Universitario de Azuero, Chitré, Panamá, 10 Departamento de Psicologia, Universidade Federal de Santa Catarina, Florianópolis, Santa Catarina, Brasil, 11 Facultad de Psicología, Universidad de Talca, Talca, Chile, 12 Departamento de Estudios Culturales, Demográficos y Políticos, Universidad de Guanajuato, Guanajuato, México, 13 Facultad de Ciencias de la Conducta, Universidad Autónoma del Estado de México, Toluca, México, 14 Universidad Comunera (UCOM), Asunción, Paraguay, 15 Facultad de Humanidades y Ciencias de la Educación, Universidad Mayor de San Andrés (UMSA), La Paz, Bolivia, 16 Asociación Proyecto Aigle, Universidad del Valle de Guatemala, Ciudad de Guatemala, Guatemala, 17 Hospital Pediátrico Docente "Borrás—Marfán", La Habana, Cuba, 18 Facultad de Psicología, Universidad de La Habana, La Habana, Cuba, 19 Universidad Pedagógica Experimental Libertador, Instituto de Mejoramiento Profesional del Magisterio, Caracas, Venezuela, 20 Fundación para el Desarrollo de la Psicología en el Caribe (FUNDEPSIC), Santo Domingo, República Dominicana, 21 Universidad Pedagógica Nacional Francisco Morazán, Tegucigalpa, Honduras, 22 Facultad de Psicología, Universidad de la República de Montevideo, Montevideo, Uruguay, 23 Fundación Pro-Educación de El Salvador (FUNPRES), San Salvador, El Salvador, 24 Facultad de Psicología, Universidad de Buenos Aires (UBA), Consejo Nacional de Investigaciones Científicas y Técnicas (CONICET), Buenos Aires, Argentina, 25 University of Mississippi, Oxford, MS, United States of America

* lucas.gagogalvagno@hotmail.com

**Data Availability Statement:** The data that support the findings of this study are openly available at:

## Abstract

Children's screen use is ubiquitous, with toddlers in particular demonstrating increases after the pandemic and negative associations with cognitive abilities. Thus, the objective of this cross-cultural study was to broaden and deepen existing results by describing Latin American toddlers' screen use and its association with parental reports of language skills, developmental milestones, and sociodemographic variables. A sample of 1878 toddlers from 12

https://osf.io/awh4q/?view_only=3a445c2ebf75469d896a0d54b66db1b3.

**Funding:** This work has been supported by the National Council of Scientific and Technical Research (CONICET, Lucas G. Gago-Galvagno Postdoctoral Fellow, 2023-2025), Ministry of Science, Technology and Innovation (MINCyT, Argentina; PICT-2021-GRFTI-00530), Open Interamerican University (UAI, 03187, 2023-2025) and Scholarships and Programs Unit of the University of Buenos Aires (UBACyT, 20020190200459BA)." The funders had no role in study design, data collection and analysis, decision to publish, or preparation of the manuscript.

**Competing interests:** The authors have declared that no competing interests exist.

to 48 months ($M.age$ = 27.55, $SD$ = 9.68, male = 933, low-SES = 945) from 19 different Latin American countries was evaluated. Parent-report measures of children's use of screens, SES, language, and developmental milestones were administered virtually or face-to-face. Results indicated that infants' screen exposure times were longer than recommended, with TV and smartphone being the most frequent screen media among them. Also, most of the screen time was shared with an adult. These results were consistent across parent-reported SES and nationality. In addition, negative and significant associations were generally found between screen time and language skills, which were in turn positively associated with shared reading times. The frequency of shared screen use with adults demonstrated positive correlations with language skills, after controlling for sociodemographic variables. Lastly, entertainment and educational content was associated with higher levels of language skills compared to music. In conclusion, this study demonstrates the importance of promoting responsible and accompanied use of screens with age-appropriate content during the first years of life across different contexts.

## Introduction

Descriptive studies show that children use screens before two years of age for an average of one hour a day or more, with background TV, TV, and cell phones as the most encountered type of screen [1–5]. This reported use goes against the recommendations of pediatric societies, which advise against screen use for children under two years and suggests shared use with parents after two years [6–8]. These recommendations are primarily based on the body of work demonstrating that screen use is typically negatively associated with cognitive and socioemotional development early in life. Researchers have also called for more work to understand how contextual considerations related to parental involvement, type of screen, and type of content may affect the impact of children's screen use on development [9, 10].

In addition, more work is needed about screen use and its links to development across different world regions and SES. The purpose of the present study is to provide data on screen use, developmental milestones, and cognitive development while also considering important contextual factors (e.g., parental sharing, media type, media content) in a novel Latin American sample.

### The importance of child, context and content in early screen use

Parent-reported use of screens from children's first three years frequently shows negative associations with socioemotional and overall cognitive skills [11, 12]. Further, longitudinal studies demonstrate that early screen use predicts lower scores in cognitive tasks later in preschool [10, 13]. For example, on a systematic review of 80 studies with children from 0 to 7 years old, more screen time was related to sleep problems, unhealthy dietary behaviors, more sedentary activities, higher levels of aggressiveness and hyperactivity, and more peer problems, among others [9]. In addition, this meta-analysis demonstrated that excessive screen time (one or two hours per day) was associated with risk of obesity (n = 9 studies) and sleep problems ($n$ = 14 studies). Similar results have been demonstrated with toddlers, with studies showing positive relationships between screen time and higher levels of behavioral problems, anxiety, and depression in a longitudinal study with 2026 children from 2 to 18 months [14] and in a parent-report with 161 toddlers [15].

Explanations for the negative relationship between screen time and development vary in what they emphasize in this relationship. Some authors [3, 16] propose a displacement hypothesis suggesting that the excessive use of screens interferes with children's physical activities and interaction with other adults, thus leading to fewer opportunities to express their emotions and flexibly respond. Additional interpretations stress that screens require that children passively engage with their environment, which is not conducive to progression in learning and development [4, 17]. Some accounts also suggest that caregivers may turn to screens as a tool to calm children, which reduces the probability that they practice the self-regulation crucial to its development (e.g., screen as babysitter) [18, 19]. Finally, some have also suggested that measures of screen time may be an indirect measure of the time children spend alone, which is an important consideration to the importance of joint engagement and attention in development [20].

## Child, context, and content considerations in the relationship between screen use and early language development

About the relationship between early screen use and language development, some studies showed that sharing screens with an adult and consuming age-appropriate educational content can be positively related to language [4, 21]. For example, a metanalysis of 42 studies of children between 0 to 12 years demonstrated that screen time exhibited generally negative associations with language skills (i.e., expressive, and receptive language), with the exception that educational content and co-viewing were positively associated with language (with low effect sizes) [3]. Considerations of the context and content of screen use is especially important during infancy and toddlerhood, as several researchers suggest that learning from screens is difficult for infants and toddlers without verbal or physical engagement from an adult—though this limitation to learning decreases with age [4, 22].

Research supports the proposal that content and context of screen use is important during infancy and toddlerhood. For instance, in a study with 85 Saudi toddlers (1- to 3-year-olds), children spent more time watching child content (and 60% of the sample reported never reading a book to infants), with passive co-viewing and background TV being the most frequent type of electronic device use. When multiple elements of screen use were entered in a regression model related to language, only context of use (active adult engagement during screen use) positively predicted toddlers' number of words understood and produced [23]. Similarly, on a scoping review of 15 studies with toddlers from 0 to 3 years old demonstrated that screen time use (reported at 1 to 3 hours on average that increased with age) showed generally negative associations with language abilities. However, these results were moderated if an adult actively accompanied its use or if the content was educational [2]. The same results were found in a sample of 117 3-year-olds Korean preschoolers, with expressive language, though it is important to note that at a descriptive level, most of this sample uses screens for less than one hour [24]. Additionally, most of the studies measured only TV use and were from North America or Europe, with few studies from other continents.

The importance of screen sharing, and screen content has been demonstrated in other studies as well. For example, some research found that parents' verbal and physical scaffolding when using novel technology related to 2- to 3-year-olds children's better verbal and non-verbal behaviors and parental reports language [4, 25]. In addition, primary caregivers used more physical and verbal interactions during the third year in comparison to the second year of life, likely because of the rapid verbal development [25]. Finally, they highlighted the importance of literacy context and reading to infants, as they found that mothers' literacy beliefs and shared reading were associated with better vocabulary outcomes [4, 25]. Literacy beliefs might be

related to the quantity and complexity of child-directed speech (and possibly, to mothers' emotional and verbal responsiveness), while shared reading provides an ideal context for pleasant and socially significant verbal interactions, thus contributing to language development [26].

Ecological examinations of screen use also suggest that consideration of content and context are important. In an ecological evaluation of families with two-year-olds, some authors measured the sound environment during a typical day and asked about screen use routines. They found that parents' tendency to use screens during childhood routines, as well as the time children spent watching TV content, was negatively associated with the child's language development. In addition, parents' use of joint media engagement (JME) was positively associated with language development, specifically pragmatic abilities [27]. Similar results were found with parent report measures, demonstrating that literacy context, JME, and general interactions between caregivers and toddlers positively predicted infant vocabulary density and use of sentences [4, 28]. Thus, consistent results suggest it is important to consider co-viewing media with your child and the interaction around the media content as well [29]. Talking about the intention of the actors and explaining their actions and feelings are aspects of JME and could be considered specific linguistic input that would be related to language. Considering child, context, and content may also help us understand null associations between screen use and language exhibited in different studies. For example, on a study with 161 primary caregivers of toddlers from 18 to 36 months using parent reports of screen usage and language development. They found low effect size correlations, and no contributions of touchscreen time on language [15]. It is possible that more detail is needed to better understand how some instances of screen use could be helpful (e.g., shared engagement, educational content) while others may be detrimental (e.g., background TV or age-inappropriate content) to language development.

## Child, context, and content considerations in the relationship between screen use and early cognitive and motor developmental milestones

While the literature surrounding the link between screen use and early language development is robust, examinations of the link between screen use and cognitive and motor developmental milestones is relatively sparse, somewhat contradictory, and typically relies on older samples. Studies that focus on milestones typically examine the age at which children achieve important cognitive and physical skills in development (e.g., age that children said first word or took the first step). The studies that do examine the link between screen use and cognitive and developmental milestones in infancy and toddlerhood seem to suggest mixed results. For example, when parents reported that toddlers ($n$ = 253, 2 to 48 months) started using screens later in life, earlier acquisition of motor and language milestones were reported [30]. Other results did not find associations between time of touchscreen use, fine and gross motor, and language milestones in 715 retrospectively parent reports of children from 6 to 36 months. Only the active scrolling with touchscreens was associated with earlier fine motor development milestones acquisition [31]. Similar results were found with 114 parent reports of children from 12 to 36 months, measuring different types of screen use and motor and language milestones. No relations between time use of TV, background TV, PC, Tablet and Cell Phone and the abilities were showed [20].

Also, some studies suggest that screen use may be related to motor development more generally, but results are mixed here as well. For example, on an intervention with 80 preschoolers that received an educational program with tablets, lower levels of fine motor and manual dexterity for those children in the intervention were showed. Also, the post-test scale score of fine motor precision, dexterity and integration was lower to the pretest intervention group, and children in the control group showed increasing scores in the three physical skills [32].

In contrast, parent's reports of children's (*n* = 117 Korean preschoolers) smart devices frequency (but not time) were positively associated with fine motor development, but only in three-year-olds and not at four and five years [20]. Also significant evidence for a positive relationship between screen use and motor development was found, demonstrating that children (*n* = 78, 24 to 48 months) who used tablets had better fine motor skills compared to those who did not use tablets. Nevertheless, the size effect was low, the screen was used with an adult in most cases, and for this sample the time of use was the recommended by the pediatric associations [33]. Thus, more attention to the child, content, and context may be particularly useful in understanding the link between screen use and early cognitive and motor milestones in infants and toddlers.

Clearly, more work examining the link between cognitive and motor milestones in the first few years of life would be useful. It is possible that, like language, the impact of screen use on developmental milestones depends on the type of screen and content; however, this is considered less when examining screen use's relations to early cognitive and developmental milestones.

## Children's developmental SES context and screen use in Latin America

One factor related to the child that may be important to understanding the impact of screen use on development is the environmental context of children. Factors related to SES (i.e., income, education, occupational prestige, and place of residence) have been the most studied. SES has been examined in relation to use of media and books at home and a robust association between early reading and SES has been established. For example, in a sample of 127,000 families from different countries with children under 5, only less than one third of the sample read books to their children, and there were positive associations with these adults' behavior and their countries' Human Development Index [34]. Also in a sample of 97,731 families from developed countries with children aged 36 to 59 months, families from low-SES backgrounds did not have books at home and children from mid-to-high SES had in general 3 or more children's books [35].

Children's contextual factors are important to consider in Latin American (LATAM) samples. Regarding the region, LATAM is going through a general period of economic crisis, since after the COVID-19 pandemic the poverty rate grew to 32.1% and extreme poverty to 13.1% [36]. In addition, annual inflation levels are high for most countries in the region (on average 14%) and present high levels of social inequality (Gini index = 0.458) [37]. On the other hand, although most pediatric associations provide uniform recommendations regarding the use of screens (i.e., zero use for children under two years of age and one hour accompanied by adults for preschoolers), they may not be considered plausible given the cultural differences that could exist between the different LATAM regions. For example, some results show in vulnerable environments the time spent on screens increases, perhaps because of the lack of access to educational resources or to information about screen use consequences [19, 20].

Thus, it is possible that the differences in the SES and other contextual factors (e.g., parenting styles) across a large sample of LATAM countries could reveal different patterns of screen use. For example, Bolivia, Haiti, and Honduras typically report the lowest SES levels in the region, considering the gross domestic product (GDP), the human development index (IDH) and the poverty rate [38]. This could be related to more screen time use, as previously said. Furthermore, within the same context some of the countries have more collectivist and extended family perspectives (e.g., Venezuela, Guatemala, and Peru), compared to other more individualistic and liberal (e.g., Argentina, Mexico, and Brazil) [39, 40], which is associated with different forms of parenting. Collectivist societies promote interdependence and

authoritarian parenting styles, and individualistic societies promote autonomy [40], which usually leads to more authoritarian and authoritative parenting styles respectively [41]. The regionally contextual aspects such as SES and parenting styles that vary across LATAM [42, 43], could be related to aspects that could limit the conditions of infant's educational resources access. However, there are no studies to date examining potential differences in screen use— let alone potential differential relationships to language and developmental milestones—across a large LATAM sample.

## The current study

The development of early language and motor skills builds an essential foundation for later development, and developmentalists need to consider how screen use relates to these developments while also considering children's broader context, and the type and content of media use. At present, research on screen use and early development in infants and toddlers elicits several important considerations: 1) negative associations between toddlers screen use time and language and motor skills are common though not definitive, 2) the use of new technologies and social vulnerability have increased substantially in recent years in LATAM and the world respectively, 3) there are no descriptive and associative data on the use of screens in a sample of diverse LATAM countries, and 4) empirical studies with novel and replicated results in a diverse and non-WEIRD sample is important to understanding screen use in a global setting. In the present study, we examine screen time with consideration of the context and content across a broad and diverse context with infants and toddlers from 19 Latin American countries. We focus on children from 12 to 48 months given the smaller number of studies, the importance of cognitive development, and the recommendations of low screen use in this age range.

We expect that infants will consume more screen time than recommended by international pediatric associations (i.e., more than 1 hour daily) across all LATAM countries and that the increased use of electronic devices will be related to lower levels of language and motor skills. Further, we hypothesize that sharing screens with an adult will positively correlate to language and motor skills, and finally, that families belonging to vulnerable socioeconomic backgrounds will show greater use of screens with their infants.

## Method

### Participants

Sampling was non-probabilistic and intentional. The children native language was Spanish (except in Brazil). Children included in the study did not present serious illnesses, chronic use of medications, headaches, or seizures based on parent reports of children's clinical history prior to evaluation. Children who were born preterm were not excluded from the final sample because there were no differences between them and full-term children regarding language and motor variables ($p > .05$).

The total sample included 2271 primary caregivers of children aged 12 to 48 months. 169 children were excluded for the final sample because they were older than 48 months, 113 were excluded for being younger than 12 months, and 111 were excluded for presenting atypical development. The final sample considered a total of 1878 primary caregivers of children between 12 and 48 months ($M$age = 27.55 months, $SD$ = 9.68, male = 933, low-SES = 945, attended daycare center = 844), from 19 LATAM countries: Argentina ($n$ = 185), Perú ($n$ = 525), Brazil ($n$ = 91), Chile ($n$ = 87), Nicaragua ($n$ = 107), Uruguay ($n$ = 12), Mexico ($n$ = 94), Venezuela ($n$ = 46), Colombia ($n$ = 84), Bolivia ($n$ = 71), Paraguay ($n$ = 86), Honduras

($n$ = 42), Ecuador ($n$ = 101), Dominican Republic ($n$ = 37), Guatemala ($n$ = 43), Panama ($n$ = 104), Cuba ($n$ = 37), Costa Rica ($n$ = 97) and El Salvador ($n$ = 29).

In terms of education and occupation, 12.9% and 16.9% of mothers and fathers didn't finish secondary school respectively, and approximately 20% of each country's subsample had completed university studies. 44% of mothers and 28.5% of fathers were unemployed, operators, or had informal jobs. 42% approximately were employees, technical or professionals, being these two categories 33% of the sample.

## Measures

**Permanent Household Survey [44].** Caregivers reported on children nationality, age (in months), gender, attendance to educational institutions (1. No, 2. Yes, public institution, 3. Yes, private institution), as well as adults' educational level (0. No studies to 10. complete post-graduate), profession (0. Unemployed, 1. Housekeeper, 2. Not qualified, 3. Operator, 4. Employee, 5. Technician, 6. Professional), overcrowding (quantity of rooms per person), and access to basic needs (3/4 meals per day, access to health and education).

Children's context were classified as unsatisfied basic needs group (UBN) if one of the following criteria were met: they lived in a precarious settlement (e.g., a non-permanent dwelling often constructed from improvised materials that may lack infrastructure related to water, electricity, and sanitation), the house had no bathroom, the house had no access to mains water, it was overcrowded (three or more people per room), elementary school-aged children in the household were not attending school, the parents in the house did not have a primary school education, and the family have no access to food, education or health.

**Children screen and book exposure.** Information from caregivers reports were collected on the following subjects: a) type of touch screens, including TV, background TV, cell phones and Tablet; b) time of screens use, considering how many minutes in a typical day children used each of these devices (free responde and "I do not possess" option); c) books exposure time (how many minutes children use interacting with a book); d) how often an adult in the family shares the screen with the toddler (1. Never, 2. Almost never, 3. Sometimes, 4. Almost always, 5. Always, 6. Does not use, 7. we do not have that device); and e) the content they mainly consumed (i.e., music, educative, entertainment).

**Developmental milestones [31].** Caregivers were asked to report on milestones by stating "At what age did the infant. . ." achieve several different critical motor and language milestones. For motor milestones caregivers were asked what age infants: "Sat without support" "Walked independently", "Picked up a small object with a clamp, that is, with their thumb and forefinger", and "Stacked at least three small blocks or other small objects". For language milestones caregivers were asked what age infants "Said their first word", "Said two or more words together", and "Made a whole sentence, meaningful". Caregiver replied to each milestone question using a 6-point scale, with the following age ranges: 0. Between 0 to 5 months, 1. Between 6 and 11 months, 2. Between 12 and 18 months, 3. Between 19 and 25 months, 4. Between 26 and 36 months, 5. Still not performed. An overall score for motor and language developmental milestones was calculated by averaging together scores from each development milestone domain. Higher scores indicated later achievement of the motor and language milestone. For this sample, McDonald's Omega approached was in the acceptable range: .64 for motor milestones and .77 for language milestones.

**Communicative Development Inventory Form II (CDI) [45].** This questionnaire inventory evaluates the development of language in children through the reporting of a significant caregiver. There were two inventories. Part 1 (lexical density) measures children's use of words. It includes a vocabulary list of 699 words divided in 23 semantic categories. Parents

reported the number of words their children know. Part 2 (sentence use) inquired about the way in which the infant uses language, specifically about the evocation of past and future events, places or people that are not present, detaching language from its immediate context (symbolic competence). Five questions with examples were asked with 3 options each (0. Not yet, 1. Sometimes, 2. Many times). The sentence use scale had a range of 0 to 10 points based on the following questions: Does the child talk about past situations? 2) Does the child talk about objects or people that are not present? 3) Does the child talk about things that are going to happen in the future? 4) Does the child understand when they are asked to bring something from another place? 5) When pointing to or grasping an object, does the infant say the name of the person to whom the object belongs even though that person is not present? For this sample, McDonald's Omega was .98 for lexical density and .84 for sentence use.

## Procedure

Surveys were conducted online ($n$ = 1648) and face-to-face ($n$ = 230). The objectives of the study were explained to all the guardians, who agreed to participate through a written informed consent (primarily endorsed by the Responsible Conduct Committee of the University of Buenos Aires, and the Universities and Research Institutes of each country). The research was anonymous, confidential, and voluntary to avoid bias in responses. No families received financial compensation, and all completed the scales individually at home or in educational institutions. The email address of the responsible researcher of each country was available to the participants to contact if there were questions or concerns.

All questionnaires were administered in the same order for all countries: First medical history questions, then the Permanent Household Survey, the questionnaire on the use of screens, questionnaire on developmental milestones, and finally the CDI. The data was collected from August 2021 to March 2023.

## Analytic plan

The Jamovi software version 2.3 was used [46]. Because Shapiro-Wilk tests showed abnormal distribution in all variables, non-parametric statistics were used. Given the purpose of our study was to provide data on screen use, developmental milestones, and language development that also consider important SES contextual factors, we conducted our analyses in several steps.

First, to provide data on screen use, language and motor milestones, and relevant SES contextual factors, we examined the descriptive statistics across the complete sample. Next, to better understand how contextual factors related to screen use, we conducted Spearman Rho's partial correlations between screen time use and sharing with an adult with language, motor, and SES variables, controlling for age, gender, parents' education and daycare assistance (because of the significant association with the variables).

Also, Kruskal-Wallis H and two-to-two Dwass-Steel-Critchlow-Fligner test were used for multiple comparisons of toddler's cognition considering type of content, screen use and sharing with, and also adult time regarding the different countries' groups.

Finally, we considered multiple factors together by including variables that had significant associations with screen use in three hierarchical regression models, taking screen use (i.e., time and sharing) as predictor, and children's language and milestones as outcomes. Independence of error assumption was met for the three models ($1.57<$ Durbin-Watson$< 1.88$). Variance inflation factors indicated that multicollinearity was not a concern in any of the models ($1.02<$ VIFs$< 1.25$). The data that support the findings of this study are openly available at: https://osf.io/awh4q/?view_only=3a445c2ebf75469d896a0d54b66db1b3

## Results

### Descriptive statistics for screen use

Descriptive statistics (Table 1) showed that TV and background TV were the most consumed screens for infants, with an average use of approximately 60 and 90 minutes per day, respectively. Only 172 infants (9.12%) from the total sample didn't use TV. Cell phones were the second most used screen, with 30 minutes on average per day and only 611 infants (32.63%) not using this screen type. However, the asymmetry for these variables was high (asymmetry > 2, [47]), so in general the use was low in most of the infants. PC and Tablet were mostly not used in this sample, with only 432 infants using a PC (23%) and 495 of children using a Tablet (26.35%). Also, for all the screens, on average the adult reported that they tend to share screen use with their infant between sometimes to always (Md = 4, Mo = 5). Finally, infants that used screens to watch specific content (*n* = 1280), mainly consumed entertainment content (59.9%), followed by music (28.7%) and finally educational content (15.3%).

### Partial correlations between language, developmental milestones, screens and books use, and UBN

Correlations generally showed that more screen use time was related to lower scores for language and motor development. Higher caregiver reports of children's time exposed to background TV and PC were related to lower reports of lexical density and sentence use. In addition, higher caregiver reports of TV time use were related to caregiver reports of later language milestones. However, it is important to note that not all reports of screen time use were related to lower language and motor abilities. Higher caregiver reports of tablet use were

**Table 1. Descriptive statistics on the use of cognitive abilities, screens and books, time sharing, and sociodemographic variables.**

| | N | Mean | Median | Mode | SD | Min | Max | Asymmetry | | Kurtosis | | Shapiro-Wilk | |
|---|---|---|---|---|---|---|---|---|---|---|---|---|---|
| | | | | | | | | Asym. | SE | Kurtosis | SE | W | P |
| Lexical Density | 1878 | 232.524 | 151.00 | 23.00 | 209.976 | 0 | 713 | 0.878 | 0.0565 | -0.5610 | 0.113 | 0.853 | < .001 |
| Use of Sentences | 1878 | 9.538 | 10.00 | 10.00 | 3.910 | 0 | 18 | -0.625 | 0.0565 | 0.0249 | 0.113 | 0.944 | < .001 |
| Motor Milestones | 1878 | 9.001 | 9.00 | 8.00 | 2.960 | 0 | 28 | 2.125 | 0.0565 | 9.9589 | 0.113 | 0.840 | < .001 |
| Langauge Milestones | 1878 | 9.509 | 9.00 | 8.00 | 3.777 | 0 | 21 | 0.781 | 0.0565 | 0.4592 | 0.113 | 0.945 | < .001 |
| TV use | 1795 | 59.389 | 30 | 60.00 | 71.374 | 0 | 600 | 2.201 | 0.0578 | 7.3959 | 0.115 | 0.770 | < .001 |
| Background TV | 1818 | 90.189 | 60.00 | 0.00 | 131.143 | 0.00 | 780 | 2.739 | 0.0574 | 8.8578 | 0.115 | 0.672 | < .001 |
| Cell Phone use | 1829 | 25.294 | 5.00 | 0.00 | 43.867 | 0.00 | 480 | 3.340 | 0.0572 | 16.7069 | 0.114 | 0.613 | < .001 |
| PC use | 1797 | 5.046 | 0.00 | 0.00 | 25.141 | 0.00 | 540 | 9.939 | 0.0577 | 146.7462 | 0.115 | 0.201 | < .001 |
| Tablet use | 1803 | 7.390 | 0 | 0.00 | 26.426 | 0 | 480 | 6.953 | 0.0576 | 77.9430 | 0.115 | 0.307 | < .001 |
| Books use | 1796 | 12.515 | 2.00 | 0.00 | 21.287 | 0 | 360 | 5.334 | 0.0578 | 58.3536 | 0.115 | 0.583 | < .001 |
| Share TV | 1706 | 4.142 | 4.00 | 5.00 | 0.929 | 1 | 5 | -1.063 | 0.0593 | 0.9780 | 0.118 | 0.803 | < .001 |
| Share Cell Phone | 1267 | 4.066 | 4 | 5.00 | 1.137 | 1 | 5 | -1.219 | 0.0687 | 0.7549 | 0.137 | 0.778 | < .001 |
| Share PC | 436 | 3.055 | 3.00 | 1.00 | 1.706 | 1 | 5 | -0.111 | 0.1169 | -1.6976 | 0.233 | 0.791 | < .001 |
| Share Tablet | 495 | 3.137 | 4 | 1.00 | 1.609 | 1 | 5 | -0.260 | 0.1098 | -1.5296 | 0.219 | 0.818 | < .001 |
| Quantity of UBN | 1878 | 0.880 | 1.00 | 0.00 | 1.131 | 0 | 5 | 1.401 | 0.0565 | 1.5480 | 0.113 | 0.763 | < .001 |
| Parents Education | 1865 | 12.209 | 12 | 8.00 | 4.849 | 0 | 20 | -0.329 | 0.0567 | -0.4504 | 0.113 | 0.970 | < .001 |
| Parents Ocupation | 1854 | 7.917 | 8.00 | 12.00 | 3.059 | 0 | 12 | -0.246 | 0.0568 | -0.5652 | 0.114 | 0.928 | < .001 |
| Toddlers Age (in months) | 1878 | 27.556 | 27.00 | 36.00 | 9.680 | 12 | 48 | 0.222 | 0.0565 | -0.7297 | 0.113 | 0.958 | < .001 |

Note: n = 1878. Child use: child media exposure in minutes. Share time: Likert scale. SD standard deviation, Min minimum, Max maximum, SE standard error, TV television, PC personal computer, UBN unsatisfied basic needs.

related to higher sentence use scores. In addition, caregiver reports of cell phone, PC and tablet time use were related to caregiver reporting that language milestones were met earlier. Also, higher caregiver reports of time using books with children was related to higher scores on both lexical density and sentence use.

Also, positive associations were found between sharing media (TV and cell phone) and language variables. For instance, higher caregiver reports of sharing TV were related to higher lexical density, sentence use scores, and earlier language milestones. Higher caregiver reports of sharing cell phones were related to higher sentence use scores. However, it is important to note that for all the correlations results, the effect sizes were low ($-.021< Rho< .133$). No relations were found between motor developmental milestones and screen use and share ($p > .05$).

Finally, there were negative correlations between UBN and Tablet screen use and sharing. When families present more UBN indicators, Tablet time use, sharing the Tablet with an adult, and infant's books use tend to decrease. Partial correlations for all variables are reported in Table 2.

## Comparisons between lexical density, sentence use and language developmental milestones considering screens type of content

There were differences between type of content and lexical density, sentence use and language developmental milestones. The group that consumed more educational content had higher scores of lexical density ($X^2 = 44.03$, $p< .001$, $\varepsilon^2 = .034$). The one that consumes more entertainment content present higher reported scores of sentence use ($X^2 = 69.14$, $p< .001$, $\varepsilon^2 = .054$) and earlier development of language developmental milestones ($X^2 = 29.13$, $p< .001$, $\varepsilon^2 = .034$). Also, significant results were found when comparing entertainment and educative content with music content, being this the one associated with less motor and language reported levels.

## Comparisons between screen use and share time considering countries

Finally, there were differences between countries in all the variables of time and sharing screen, but the effect sizes were low for all comparisons ($34.4<X^2 <168.2$, p< .01, .04< $\varepsilon^2< .09$). These differences remained after controlling for age, gender, parents' education and daycare assistance.

## Contribution of screen use and share to language development

Tables 3–5 summarizes the results of the regressions for lexical density ($R^2 = .270$, $F = 30.1$, $p< .001$), sentence use ($R^2 = .223$, $F = 23.4$, $p< .001$) and language developmental milestones ($R^2 = .101$, $F = 9.16$, $p< .001$) respectively. For the three dependent variables, the general model was significant. Books use and shared TV positively predicted lexical density and sentence use, being that as the time of books and sharing TV with an adult increased, the scores reported by parents of lexical density and the use of sentences were higher for this LATAM sample. Regarding language developmental milestones, the use of TV and Cell Phone predicted later parent reported acquisition of these milestones, but PC and Tablet were associated with earlier acquisition of language developmental milestones.

## Discussion

The objective of this study was to describe the type of screen use in a diverse sample of LATAM toddlers, and to associate this use with language and motor development, social

**Table 2. Partial Spearman correlation between cognitive abilities, use of screens and books, time sharing, and sociodemographic variables.**

| | 1. | 2. | 3. | 4. | 5. | 6. | 7. | 8. | 9. | 10. | 11. | 12. | 13. | 14. | 15. |
|---|---|---|---|---|---|---|---|---|---|---|---|---|---|---|---|
| 1. Lexical density | — | | | | | | | | | | | | | | |
| 2. Sentences use | 0.47 *** | — | | | | | | | | | | | | | |
| 3. Motor DM. | -0.14 *** | -0.13 *** | — | | | | | | | | | | | | |
| 4. Language DM. | -0.26 *** | -0.22 *** | 0.41 *** | — | | | | | | | | | | | |
| 5. TV use | -0.03 | -0.01 | -0.01 | 0.06 ** | — | | | | | | | | | | |
| 6. Background TV | -0.03 | -0.04 * | -0.01 | 0.08 *** | 0.30 *** | — | | | | | | | | | |
| 7. Cell Phone use | -0.01 | 0.04 | -0.02 | -0.04 * | 0.19 *** | 0.19 *** | — | | | | | | | | |
| 8. PC use | -0.05 * | 0.02 | -0.03 | -0.07 ** | -0.02 | -0.04 | 0.09 *** | — | | | | | | | |
| 9. Tablet use | 0.01 | 0.11 *** | -0.02 | -0.10 *** | -0.01 | -0.01 | 0.06 ** | 0.32 *** | — | | | | | | |
| 10. Books use | 0.15 *** | 0.13 *** | -0.04 | 0.01 | 0.02 | -0.07 ** | -0.02 | 0.05 * | 0.05 * | — | | | | | |
| 11. Share TV | 0.11 *** | 0.13 *** | -0.04 | -0.06 ** | -0.02 | -0.03 | -0.03 | -0.04 | -0.01 | 0.05 | — | | | | |
| 12. Share Cell Phone | 0.03 | 0.06 | -0.00 | -0.01 | -0.02 | -0.01 | 0.15 *** | -0.09 *** | -0.08 ** | 0.06 * | 0.47 *** | — | | | |
| 13. Sahare PC | -0.02 | 0.02 | 0.02 | -0.01 | 0.05 | 0.01 | 0.09 | 0.62 *** | 0.12 * | 0.11 * | 0.21 *** | 0.43 *** | — | | |
| 14. Share Tablet | -0.01 | 0.06 | 0.04 | 0.02 | -0.02 | -0.02 | 0.02 | 0.11 * | 0.52 *** | 0.09 * | 0.38 *** | 0.53 *** | 0.66 *** | — | |
| 15. UBN Quantity | -0.09 *** | -0.09 *** | 0.01 | -0.04 | -0.03 | 0.01 | 0.02 | 0.02 | -0.06 * | -0.10 *** | -0.03 | -0.01 | 0.01 | -0.09 * | — |

Note. Control for 'Parents Education', 'Daycare asistance', 'Toddlers gender', and 'Toddlers age (in months)'. DM developmental milestones, TV television, PC personal computer, UBN unsatisfied basic needs

* p < .05

** p < .01

*** p < .001

**Table 3. Hierarchical linear regression model of toddler's lexical density.**

| Predictor | Estimator | SE | t | p | β | $R^2$ | df1 | df2 |
|---|---|---|---|---|---|---|---|---|
| Constant | -252.7939 | 42.4588 | -5.9539 | < .001 | | .213 | 4 | 292 |
| Toddlers Gender | 45.4397 | 11.1747 | 4.0663 | < .001 | 0.10783 | | | |
| Toddlers Age (in months) | 11.3923 | 0.6484 | 17.5686 | < .001 | 0.49961 | | | |
| Daycare assistence | -12.3217 | 6.9376 | -1.7761 | 0.076 | -0.04990 | | | |
| Parents Education | 4.1433 | 1.2494 | 3.3161 | < .001 | 0.09167 | | | |
| UBN Quantity | -8.3268 | 5.3222 | -1.5645 | 0.118 | -0.04286 | | | |
| TV use | 0.0171 | 0.0790 | 0.2169 | 0.828 | 0.00604 | .243 | 10 | 286 |
| Background TV | 0.0109 | 0.0424 | 0.2570 | 0.797 | 0.00709 | | | |
| Cell Phone use | -0.1244 | 0.1359 | -0.9150 | 0.360 | -0.02616 | | | |
| PC use | -0.0320 | 0.2301 | -0.1391 | 0.889 | -0.00383 | | | |
| Tablet use | 0.3886 | 0.2035 | 1.9095 | 0.056 | 0.05297 | | | |
| Books use | 0.6549 | 0.2460 | 2.6627 | 0.008 | 0.07136 | | | |
| Share TV | 19.1342 | 6.6769 | 2.8657 | 0.004 | 0.08448 | .266 | 14 | 282 |
| Share Cell Phone | 0.3376 | 5.3557 | 0.0630 | 0.950 | 0.00186 | | | |

Note: Child use: child media exposure in minutes. Share time: Likert scale. SE standard error, TV television, PC personal computer, UBN unsatisfied basic needs.

vulnerability. Results showed that, on average, TV and entertainment content were the most consumed screen, and the time children spent using screens was more than the recommendations of pediatrics associations for this age range. The exposure to TV, background TV and cell phones were in general negatively associated with language and motor abilities (after controlling for child age, gender, daycare, and caregiver education). Results for PC and tablet use were mixed, and books use and sharing some media between adults and toddlers were positively related to the language abilities scores. Together, results indicate that the influence of screen use on language and motor development is complex—with some evidence replicating

**Table 4. Hierarchical linear regression model of toddler's sentence use.**

| Predictor | Estimator | SE | t | p | β | $R^2$ | df1 | df2 |
|---|---|---|---|---|---|---|---|---|
| Constant | 3.65550 | 0.65758 | 5.5590 | < .001 | | .150 | 4 | 292 |
| Toddlers Gender | 0.46544 | 0.17307 | 2.6893 | 0.007 | 0.07357 | | | |
| Toddlers Age (in months) | 0.14474 | 0.01004 | 14.4127 | < .001 | 0.42284 | | | |
| Daycare assistence | -0.00969 | 0.10745 | -0.0901 | 0.928 | -0.00261 | | | |
| Parents Education | 0.02049 | 0.01935 | 1.0591 | 0.290 | 0.03020 | | | |
| UBN Quantity | -0.21125 | 0.08243 | -2.5629 | 0.011 | -0.07244 | | | |
| TV use | -4.25e−4 | 0.00122 | -0.3472 | 0.729 | -0.00997 | .168 | 10 | 286 |
| Background TV | -3.20e−4 | 6.57e-4 | -0.4872 | 0.626 | -0.01387 | | | |
| Cell Phone use | 0.00107 | 0.00210 | 0.5088 | 0.611 | 0.01501 | | | |
| PC use | 0.00556 | 0.00356 | 1.5600 | 0.119 | 0.04432 | | | |
| Tablet use | 0.00537 | 0.00315 | 1.7052 | 0.088 | 0.04880 | | | |
| Books use | 0.01367 | 0.00381 | 3.5885 | < .001 | 0.09922 | | | |
| Share TV | 0.35444 | 0.10341 | 3.4276 | < .001 | 0.10424 | .238 | 14 | 282 |
| Share Cell Phone | 0.04525 | 0.08295 | 0.5456 | 0.585 | 0.01658 | | | |

Note: Child use: child media exposure in minutes. Share time: Likert scale. SE standard error, TV television, PC personal computer, UBN unsatisfied basic needs.

**Table 5. Hierarchical linear regression model of toddler's language development milestones.**

| Predictor | Estimator | SE | t | p | β | R² | df1 | df2 |
|---|---|---|---|---|---|---|---|---|
| Constant | 11.76939 | 0.75460 | 15.597 | < .001 | | .266 | 4 | 292 |
| Toddlers Gender | -0.80031 | 0.19860 | -4.030 | < .001 | -0.11859 | | | |
| Toddlers Age (in months) | -0.08063 | 0.01152 | -6.996 | < .001 | -0.22081 | | | |
| Daycare assistence | -0.17481 | 0.12330 | -1.418 | 0.157 | -0.04421 | | | |
| Parents Education | 0.05979 | 0.02221 | 2.693 | 0.007 | 0.08260 | | | |
| UBN Quantity | -0.01385 | 0.09459 | -0.146 | 0.884 | -0.00445 | | | |
| TV use | 0.00369 | 0.00140 | 2.631 | 0.009 | 0.08126 | .291 | 10 | 286 |
| Background TV | 0.00121 | 7.54e-4 | 1.604 | 0.109 | 0.04914 | | | |
| Cell Phone use | 0.00557 | 0.00242 | 2.305 | 0.021 | 0.07311 | | | |
| PC use | -0.00984 | 0.00409 | -2.405 | *0.016* | -0.07351 | | | |
| Tablet use | -0.00764 | 0.00362 | -2.113 | 0.035 | -0.06503 | | | |
| Books use | -0.00101 | 0.00437 | -0.232 | 0.817 | -0.00690 | | | |
| Share TV | 1.12e-4 | 0.11867 | 9.43e-4 | 0.999 | 3.09e-5 | .321 | 14 | 282 |
| Share Cell Phone | -0.03826 | 0.09518 | -0.402 | 0.688 | -0.01314 | | | |

Note: Child use: child media exposure in minutes. Share time: Likert scale. SE standard error, TV television, PC personal computer, UBN unsatisfied basic needs.

the negative relations found between screen use and early language in a large Latin American and other evidence stressing the importance of considering contextual factors when understanding screen use's impact on early development.

## Screen use in Latin America

Our descriptive results regarding early screen time are like prior work conducted across different countries. For instance, we showed that like other studies [3, 4, 7, 48], young children from Latin America used screens more than the recommended time, independent of SES. The average screen time was more than an hour per day for TV and background TV, like most other countries [3, 23], suggesting the importance of creating interventions that start early in life focused on responsible media use. There are several potential reasons why we may see high screen use in LATAM regions. For one, to the author's knowledge, there are no public educational programs focused on screen use in the first years of life in the LATAM region. Although some guides of responsible screen use have begun to be developed by region [49], this is not widespread. It is also important to consider that although screen use was above recommended levels, it was not consistent across all forms of media. TV and background TV were the most used screens, perhaps because less direct involvement is needed, so there is less probability of breaking the device. TV also provides a lot of age-targeted content options (like cell phones), and streaming allows families to actively select content, which may increase the probability of its use.

## Relations between screen use, language development, and motor development

In terms of relations between screen use and language development, TV use showed the most negative relations to language development, while Tablet, cell phone, and PC had mixed results. These results go hand in hand with other studies with infants and toddlers [2, 18] that highlight the idea that the TV is a more passive consumption device, since it may not involve any type of physical (e.g., scrolling, touching) or verbal interaction with an adult. Furthermore,

when TV is on in the background, it could hinder the communication channels between adults and children, decreasing the chances of generating meaningful interactions [11, 21]. On the other hand, the positive results with the reading time to infants' language could be because involves active adult participation and can lead to a greater number of reciprocal interactions —which research has shown to be important to language development [30, 34].

It is also important to note that while we did find negative associations between screen use and language development, this was not for all types of screens and effect sizes were generally low. These low effect sizes could be because screens can be used in a variety of ways that may not be captured by an estimate of general screen use time. For instance, parents are usually present when toddlers use screens [33] and the content is predominantly entertainment. Parents may be present because they use screens as a period of relaxation and exchanges with infants, or in the case of cell phones, tablets, or PCs (which were less used) because they are fragile devices. In this sense, it is important to note that only a few families used PCs and Tablets, and that their use was associated with a higher SES—which also may be an important consideration in understanding how screen time influences development. Thus, measures of screen use should likely consider other contextual factors related to the child, context, and content.

Our results support the idea that child, context, and content are necessary considerations when attempting to understand the impact of screen use on language and motor development. For one, we found that sharing screens between the child and the adult was associated with better performance in the reports of lexical density, sentences, and language milestones (though only for TV and cell phones). Therefore, it could be affirmed that the time of use would not be by itself the one that would have a negative contribution in the development of the language, but its use alone and passively [5, 9]. Regarding the associations with UBN quantity, only a few associations were found only related to the use of tablets and books. In the case of tablets, in the LATAM context their consumption is not very high, and they are usually only owned by families of medium to high SES backgrounds [19, 20]. Regarding books, previous studies also showed that families from vulnerable areas have fewer books at home and spend less time reading with infants [20, 35], perhaps because they are unaware of the benefits for early development, or because they do not have time and a suitable place in the home to carry out this shared activity. The absence of associations between the time of use, the shared use and the type of content with SES could be because its use is ubiquitous and the offer of content during early childhood is varied [5]. This would go hand in hand with the homogeneous results obtained by country, although statistically significant differences were found between the different countries after controlling for children's age, gender, parents' education, and day-care assistance.

Finally, we did not find a relation between screen time and achievement of later motor milestones as we hypothesized. This may not be entirely surprising as the results in this area are sparse, mixed, and typically with older samples. Thus, the lack of associations with motor development milestones early in life could be because the impact of screens on this ability would occur in later years, as shown by other studies [30, 31]. It would be necessary in future studies to evaluate the type of scaffolding (i.e., verbal, or physical) that adults perform during the use of shared screens [4, 9].

## Conclusion

This work shows that it is important to continue studying the impact of screen devices on child development, from early stages of life; and the role of the primary caregiver to scaffold their use and promote the consumption of age-appropriate content. The results of this study

replicate those of previous research [2, 3, 7, 18], in a diverse sample of LATAM toddlers, which is important to sustain the creation of public policies to reinforce the responsible use of screens, since they are being used with some independence from the SES and nationality. Our work suggests that understanding the varied factors that surround children's screen use in addition to time (e.g., content, shared use, age, available resources) will be important to creating more comprehensive policies related to how families should navigate screen use—which is only poised to increase and become more complex. More research should be carried out in this issue, with experimental designs to control variables and isolate their effects, to know the particularities of their effects.

Despite these findings, the present study has a set of limitations. On the one hand, the sampling carried out was non-probabilistic, although the sample size was relatively large. Therefore, although the sample is not representative of the region, it would have a good level of statistical power. On the other hand, the evaluation of the use of screens and of children's cognition was carried out with parental reports, which could underestimate the time of use and overestimate children's cognitive abilities due to social desirability [4]. Lastly, the research was cross-sectional, which does not allow monitoring the use of screens and development trajectories over time. For future studies, it would be desirable to carry out a probabilistic sampling, with direct behavioral measurement of cognitive abilities, and the well-being reports of the cell phone and home TV could be used, to see how long the family uses these devices, although it would not be discriminated by the time of use of the infant. Also, being able to collect a larger number of samples per country, and with homogeneous quantity, would allow for more sophisticated analysis of structural equations to evaluate the factorial invariance of the theoretical model obtained in this study. Finally, a longitudinal and experimental study would lead to more finished results for the implementation of specific public policies in educational environments.

## Supporting information

**S1 Checklist. Human participants research checklist.**
(DOCX)

**S1 Table. Country-level sample data.**
(DOCX)

**S1 File. Inclusivity in global research.**
(DOCX)

## Acknowledgments

We are deeply grateful to Susana Stoisa, Lely Galvagno, and caregivers for their valuable cooperation. The authors declare no conflicts of interest about the funding source for this study.

## Author Contributions

**Conceptualization:** Lucas G. Gago-Galvagno, Angel M. Elgier.

**Data curation:** Lucas G. Gago-Galvagno.

**Formal analysis:** Lucas G. Gago-Galvagno, Angel J. Tabullo, Stephanie E. Miller.

**Funding acquisition:** Lucas G. Gago-Galvagno, Angel M. Elgier.

**Investigation:** Lucas G. Gago-Galvagno, Edson J. Huaire-Inacio, Angela M. Herrera-Alvarez, Carmen Zambrano-Villalba, Frania R. López, Emmanuel Herrera-González, Olivia Morán-

Núñez, Neyra J. Ochoa-Vega, Chrissie Ferreira de Carvalho, Rosario Spencer, Juan José Giraldo-Huertas, Perla del Carpio, Erika Robles, Carla Fernández, Silvia Requena, Pilar Rodríguez, Yoysy Rondón, Alexis L. Ruiz, Ada Tibisay-Echenique, Gris Hidalgo, Carlos R. Hernández, Mirna Lacayo, Esther Angeriz, Gabriela Etchebehere, Fernando José Mena, Delia Magaña de Ávila, Susana C. Azzollini, Stephanie E. Miller.

**Methodology:** Lucas G. Gago-Galvagno, Susana C. Azzollini, Stephanie E. Miller.

**Project administration:** Lucas G. Gago-Galvagno, Angel M. Elgier, Susana C. Azzollini.

**Resources:** Angel M. Elgier.

**Software:** Lucas G. Gago-Galvagno.

**Supervision:** Angel J. Tabullo, Susana C. Azzollini, Stephanie E. Miller.

**Validation:** Stephanie E. Miller.

**Visualization:** Lucas G. Gago-Galvagno.

**Writing – original draft:** Lucas G. Gago-Galvagno, Edson J. Huaire-Inacio, Angela M. Herrera-Alvarez, Carmen Zambrano-Villalba, Frania R. López, Emmanuel Herrera-González, Olivia Morán-Núñez, Neyra J. Ochoa-Vega, Chrissie Ferreira de Carvalho, Rosario Spencer, Juan José Giraldo-Huertas, Perla del Carpio, Erika Robles, Carla Fernández, Silvia Requena, Pilar Rodríguez, Yoysy Rondón, Alexis L. Ruiz, Ada Tibisay-Echenique, Gris Hidalgo, Carlos R. Hernández, Mirna Lacayo, Esther Angeriz, Gabriela Etchebehere, Fernando José Mena, Delia Magaña de Ávila, Susana C. Azzollini, Stephanie E. Miller.

**Writing – review & editing:** Lucas G. Gago-Galvagno, Angel M. Elgier, Angel J. Tabullo, Edson J. Huaire-Inacio, Angela M. Herrera-Alvarez, Carmen Zambrano-Villalba, Frania R. López, Emmanuel Herrera-González, Olivia Morán-Núñez, Neyra J. Ochoa-Vega, Chrissie Ferreira de Carvalho, Rosario Spencer, Juan José Giraldo-Huertas, Perla del Carpio, Erika Robles, Carla Fernández, Silvia Requena, Pilar Rodríguez, Yoysy Rondón, Alexis L. Ruiz, Ada Tibisay-Echenique, Gris Hidalgo, Carlos R. Hernández, Mirna Lacayo, Esther Angeriz, Gabriela Etchebehere, Fernando José Mena, Delia Magaña de Ávila, Susana C. Azzollini, Stephanie E. Miller.

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
