## [Decision Letter · Decision Letter 0]

20 Sep 2024

PONE-D-24-23233Use of screens, books and adults’ interactions on toddler's language and motor skills: a cross-cultural study among 19 Latin American countries from different SESPLOS ONE

Dear Dr. Gago-Galvagno,

Thank you for submitting your manuscript to PLOS ONE. After careful consideration, we feel that it has merit but does not fully meet PLOS ONE’s publication criteria as it currently stands. Therefore, we invite you to submit a revised version of the manuscript that addresses the points raised during the review process.

We look forward to receiving your revised manuscript.

Kind regards,

Emma Duerden

Academic Editor

PLOS ONE

3. We note that your Data Availability Statement is currently as follows: [All relevant data are within the manuscript and its Supporting Information files.] Please confirm at this time whether or not your submission contains all raw data required to replicate the results of your study. Authors must share the “minimal data set” for their submission. PLOS defines the minimal data set to consist of the data required to replicate all study findings reported in the article, as well as related metadata and methods (https://journals.plos.org/plosone/s/data-availability#loc-minimal-data-set-definition). For example, authors should submit the following data: - The values behind the means, standard deviations and other measures reported; - The values used to build graphs; - The points extracted from images for analysis. Authors do not need to submit their entire data set if only a portion of the data was used in the reported study. If your submission does not contain these data, please either upload them as Supporting Information files or deposit them to a stable, public repository and provide us with the relevant URLs, DOIs, or accession numbers. For a list of recommended repositories, please see https://journals.plos.org/plosone/s/recommended-repositories. If there are ethical or legal restrictions on sharing a de-identified data set, please explain them in detail (e.g., data contain potentially sensitive information, data are owned by a third-party organization, etc.) and who has imposed them (e.g., an ethics committee). Please also provide contact information for a data access committee, ethics committee, or other institutional body to which data requests may be sent. If data are owned by a third party, please indicate how others may request data access.

Additional Editor Comments (if provided):

Reviewers' comments:

Reviewer's Responses to Questions

**Comments to the Author**

1. Is the manuscript technically sound, and do the data support the conclusions?

Reviewer #1: Yes

2. Has the statistical analysis been performed appropriately and rigorously? 

Reviewer #1: Yes

3. Have the authors made all data underlying the findings in their manuscript fully available?

Reviewer #1: Yes

4. Is the manuscript presented in an intelligible fashion and written in standard English?

Reviewer #1: Yes

5. Review Comments to the Author

Reviewer #1: Greetings.

Overall the write up is standard. However, I would suggest the following adjustments for upgrading the readability. This is an impressive endeavor and I wish for such important findings to be made public.

Introduction:

Please reduce the number of sentences. The introduction is very long. For each point of the introduction, try to limit the sentence count to 10 or fewer. Although the information provided is relevant, it can be more concise.

Methodology-Analytical Plan:

Please rewrite this part in sub-sections or paragraphs addressing each of the analyses conducted in detail. The analyses are presented in the results section; however, the details should be mentioned here. It would be best to specify which statistical analysis was conducted for what reason, the variables included, and the rationale for their inclusion. Almost all information is already in this manuscript, but listing the analyses and their background in the Analytical Plan section of the Methodology is more appropriate. In addition to "In addition, we conducted multiple group comparisons to assess the existence of differences in the

use of screens according to the gender, age, and nationality of the children." please include which tests were done too.

Result:

Please reorganize the results section to follow a more systematic approach. Remove the analysis details from this section and include them in the Analytical Plan. Ensure that tables or figures are provided for all respective results. Currently, it is difficult to discern which analysis yielded which result; this can be improved by using cross-referencing and organizing the results sequentially. For example, in the results section, one paragraph mentions:

"Partial correlations aimed to examine links between screen use (e.g., time, sharing, content)

and language and motor development in sample when controlling for child characteristics and

sociodemographic variables that relate to development. For child characteristics in the present

study, we found that age related to better language and earlier milestones while female children

get more scores in lexical density (U= 405, p= .003, r= .079) and language milestones (U= 399,

p< .001, r= .093). For sociodemographic variables we found that higher caregiver education was

related to better language and earlier milestones and access to daycare was related to more scores

in parents' reports of lexical density (X²= 73.39, p< .001, ε²= .039), use of sentences (X²= 46.95,

p< .001, ε²= .025) and earlier acquisition of language development milestones (X²= 35.56, p< .001,

ε²= .019). Thus age, gender, caregiver education, and daycare access were included as control

variables in Spearman parietal correlations to determine whether links between screen use,

language, and motor development existed when controlling for these variables. Partial correlations

for all variables are reported in Table 2."

There is no mention of where these results were calculated, nor are there any tables or figures referencing them. " For child characteristics in the present

study, we found that age related to better language and earlier milestones while female children

get more scores in lexical density (U= 405, p= .003, r= .079) and language milestones (U= 399,

p< .001, r= .093). " and "For sociodemographic variables we found that higher caregiver education was

related to better language and earlier milestones and access to daycare was related to more scores

in parents' reports of lexical density (X²= 73.39, p< .001, ε²= .039), use of sentences (X²= 46.95,

p< .001, ε²= .025) and earlier acquisition of language development milestones (X²= 35.56, p< .001,

ε²= .019). ".

In this paragraph, multiple test results are written together. The rationale for conducting these tests should be outlined in the Analytic Plan section, along with an explanation of why these results are relevant enough to be included alongside the partial correlation findings. For clarity, it would be beneficial to present each test result in separate paragraphs, enhancing comprehension for non-statistically inclined readers. Simplifying the presentation of these critical findings is necessary to ensure they are accessible and understandable to a broader audience. Please rewrite or reorganize the result section to improve the readability.

Discussion and Conclusion: No changes are required.

6. PLOS authors have the option to publish the peer review history of their article (what does this mean?). If published, this will include your full peer review and any attached files.

Reviewer #1: No

---

## [Author Response · Author response to Decision Letter 0]

25 Oct 2024

The manuscript meets PLOS ONE's style requirements.

A complete copy of PLOS’ questionnaire on inclusivity in global research was included. 

3. We note that your Data Availability Statement is currently as follows: [All relevant data are within the manuscript and its Supporting Information files.] Please confirm at this time whether or not your submission contains all raw data required to replicate the results of your study. Authors must share the “minimal data set” for their submission. PLOS defines the minimal data set to consist of the data required to replicate all study findings reported in the article, as well as related metadata and methods (https://journals.plos.org/plosone/s/data-availability#loc-minimal-data-set-definition). For example, authors should submit the following data: - The values behind the means, standard deviations and other measures reported; - The values used to build graphs; - The points extracted from images for analysis. Authors do not need to submit their entire data set if only a portion of the data was used in the reported study. If your submission does not contain these data, please either upload them as Supporting Information files or deposit them to a stable, public repository and provide us with the relevant URLs, DOIs, or accession numbers. For a list of recommended repositories, please see https://journals.plos.org/plosone/s/recommended-repositories. If there are ethical or legal restrictions on sharing a de-identified data set, please explain them in detail (e.g., data contain potentially sensitive information, data are owned by a third-party organization, etc.) and who has imposed them (e.g., an ethics committee). Please also provide contact information for a data access committee, ethics committee, or other institutional body to which data requests may be sent. If data are owned by a third party, please indicate how others may request data access.

The complete database and the analysis made are available in the manuscript, with an OSF URL. 

A full ethics statement was included in the procedure section. 

The reference list was reviewed. Thank you for the follow up and feedback. 

Reviewers' comments:

Reviewer's Responses to Questions

Comments to the Author

1. Is the manuscript technically sound, and do the data support the conclusions?

Reviewer #1: Yes

Thank you for your positive feedback. 

2. Has the statistical analysis been performed appropriately and rigorously?

Reviewer #1: Yes

Thank you for your positive feedback. 

3. Have the authors made all data underlying the findings in their manuscript fully available?

Reviewer #1: Yes

Thank you for your positive feedback. 

4. Is the manuscript presented in an intelligible fashion and written in standard English?

Reviewer #1: Yes

Thank you for your positive feedback.

5. Review Comments to the Author

Reviewer #1: Greetings.

Overall the write up is standard. However, I would suggest the following adjustments for upgrading the readability. This is an impressive endeavor and I wish for such important findings to be made public.

Introduction:

Please reduce the number of sentences. The introduction is very long. For each point of the introduction, try to limit the sentence count to 10 or fewer. Although the information provided is relevant, it can be more concise.

The introduction was reduced from 3016 to 2796 words. 

Methodology-Analytical Plan:

Please rewrite this part in sub-sections or paragraphs addressing each of the analyses conducted in detail. The analyses are presented in the results section; however, the details should be mentioned here. It would be best to specify which statistical analysis was conducted for what reason, the variables included, and the rationale for their inclusion. Almost all information is already in this manuscript, but listing the analyses and their background in the Analytical Plan section of the Methodology is more appropriate. In addition to "In addition, we conducted multiple group comparisons to assess the existence of differences in the

use of screens according to the gender, age, and nationality of the children." please include which tests were done too.

Different paragraphs with each statistical test analysis made in the manuscript were included in detail in the Analytical Plan. The new different paragraphs follow the same order as the results sections to facilitate the comprehension of the article. 

Result:

Please reorganize the results section to follow a more systematic approach. Remove the analysis details from this section and include them in the Analytical Plan. Ensure that tables or figures are provided for all respective results. Currently, it is difficult to discern which analysis yielded which result; this can be improved by using cross-referencing and organizing the results sequentially. For example, in the results section, one paragraph mentions:

"Partial correlations aimed to examine links between screen use (e.g., time, sharing, content)

and language and motor development in sample when controlling for child characteristics and

sociodemographic variables that relate to development. For child characteristics in the present

study, we found that age related to better language and earlier milestones while female children

get more scores in lexical density (U= 405, p= .003, r= .079) and language milestones (U= 399,

p< .001, r= .093). For sociodemographic variables we found that higher caregiver education was

related to better language and earlier milestones and access to daycare was related to more scores

in parents' reports of lexical density (X²= 73.39, p< .001, ε²= .039), use of sentences (X²= 46.95,

p< .001, ε²= .025) and earlier acquisition of language development milestones (X²= 35.56, p< .001,

ε²= .019). Thus age, gender, caregiver education, and daycare access were included as control

variables in Spearman parietal correlations to determine whether links between screen use,

language, and motor development existed when controlling for these variables. Partial correlations

for all variables are reported in Table 2."

There is no mention of where these results were calculated, nor are there any tables or figures referencing them. " For child characteristics in the present

study, we found that age related to better language and earlier milestones while female children

get more scores in lexical density (U= 405, p= .003, r= .079) and language milestones (U= 399,

p< .001, r= .093). " and "For sociodemographic variables we found that higher caregiver education was

related to better language and earlier milestones and access to daycare was related to more scores

in parents' reports of lexical density (X²= 73.39, p< .001, ε²= .039), use of sentences (X²= 46.95,

p< .001, ε²= .025) and earlier acquisition of language development milestones (X²= 35.56, p< .001,

ε²= .019). ".

In this paragraph, multiple test results are written together. The rationale for conducting these tests should be outlined in the Analytic Plan section, along with an explanation of why these results are relevant enough to be included alongside the partial correlation findings. For clarity, it would be beneficial to present each test result in separate paragraphs, enhancing comprehension for non-statistically inclined readers. Simplifying the presentation of these critical findings is necessary to ensure they are accessible and understandable to a broader audience. Please rewrite or reorganize the result section to improve the readability.

The results section was reorganized to follow a more systematic approach. The analysis details previously included in the results section were moved to the Analytic Plan. Additionally, each test result is now presented in separate paragraphs, with distinct subtitles to better organize the information. Finally, we simplified the content, highlighting the most important findings to enhance the article's readability and comprehension.

Discussion and Conclusion: No changes are required.

Thank you for your positive feedback. 

6. PLOS authors have the option to publish the peer review history of their article (what does this mean?). If published, this will include your full peer review and any attached files.

Do you want your identity to be public for this peer review? For information about this choice, including consent withdrawal, please see our Privacy Policy.

Reviewer #1: No

---

## [Editor Report · Decision Letter 1]

13 Nov 2024

Use of screens, books and adults’ interactions on toddler's language and motor skills: a cross-cultural study among 19 Latin American countries from different SES

PONE-D-24-23233R1

Dear Dr. Gago-Galvagno,

We’re pleased to inform you that your manuscript has been judged scientifically suitable for publication and will be formally accepted for publication once it meets all outstanding technical requirements.

Kind regards,

Emma Duerden

Academic Editor

PLOS ONE

Additional Editor Comments (optional):

The authors have addressed the recommendations in the revised version of the manuscript.
---

## [Editor Report · Acceptance letter]

29 Nov 2024

PONE-D-24-23233R1 

PLOS ONE

Dear Dr. Gago-Galvagno, 

I'm pleased to inform you that your manuscript has been deemed suitable for publication in PLOS ONE. Congratulations! Your manuscript is now being handed over to our production team.

Kind regards, 

on behalf of

Dr. Emma Duerden 

Academic Editor

PLOS ONE